# Sex-Specific Autoimmune Comorbidity Patterns in Pemphigus Vulgaris and Bullous Pemphigoid: A Bicenter Retrospective Case–Control Study

**DOI:** 10.3390/medicina61111946

**Published:** 2025-10-30

**Authors:** Özge Zorlu, Serkan Yazici, Sidar İlik, Emel Bülbül Başkan, Hülya Albayrak, Sema Aytekin

**Affiliations:** 1Department of Dermatology and Venereology, Faculty of Medicine, Tekirdağ Namık Kemal University, 59030 Tekirdağ, Turkey; drhulyaalbayrak@gmail.com (H.A.); semaaytekin@yahoo.com (S.A.); 2Department of Dermatology and Venereology, Faculty of Medicine, Bursa Uludağ University, 16059 Bursa, Turkey; serkanyazici@uludag.edu.tr (S.Y.); drsidarilik@gmail.com (S.İ.); bbemel@uludag.edu.tr (E.B.B.)

**Keywords:** pemphigus vulgaris, bullous pemphigoid, autoimmune diseases, Graves’ disease, Hashimoto thyroiditis

## Abstract

*Background and Objectives*: While pemphigus vulgaris (PV) and bullous pemphigoid (BP) have been linked to autoimmune comorbidities, the spectrum and specificity of these associations remain uncertain. We aimed to investigate the prevalence and patterns of autoimmune diseases (AIDs) in patients with PV and BP compared with age- and sex-matched controls. *Materials and Methods*: We conducted a bicenter, retrospective case–control study including 287 PV patients with 1148 matched controls and 284 BP patients with 1137 matched controls. Autoimmune comorbidities were identified through medical record review, and disease-specific as well as system-level associations between PV, BP, and AIDs were assessed. *Results*: Overall AID prevalence was lower in PV (9.4%) and BP (8.1%) than in controls (18% and 15%, respectively; *p* < 0.001 and *p* = 0.002). PV was associated with Graves’ disease (adjusted OR: 3.16, 95% CI: 1.24–8.06), especially in females. BP was associated with Hashimoto thyroiditis (adjusted OR: 2.51, 95% CI: 1.33–4.75), particularly in males. System-level analyses revealed that cutaneous and multisystem AIDs were less frequent in both PV and BP (*p* < 0.001 for each and *p* = 0.001 for each, respectively), whereas endocrine AIDs were more frequent in BP (*p* = 0.038). Thyroid antibody positivity did not differ significantly between patients and controls. Limitations include retrospective design, possible overrepresentation of cutaneous AIDs in dermatology-based controls, and lack of external validation. *Conclusions*: Our findings suggest that PV and BP may be associated with selective, sex- and phenotype-specific autoimmune comorbidity patterns rather than a generalized autoimmune burden. Further prospective studies are needed to confirm these exploratory associations and clarify their temporal relationships.

## 1. Introduction

Autoimmune bullous diseases (AIBDs) are characterized by pathogenic autoantibodies against structural proteins of the skin and mucosae, with pemphigus vulgaris (PV) and bullous pemphigoid (BP) being the most common subtypes [1].

Pemphigus vulgaris is an intraepidermal AIBD caused by autoantibodies targeting desmoglein 3 and, less frequently, desmoglein 1, leading to loss of keratinocyte adhesion (acantholysis) and mucocutaneous blister formation. It can occur at any age, but most commonly affects individuals between 45 and 65 years at diagnosis [2,3]. In contrast, BP is a subepidermal AIBD in which autoantibodies against the hemidesmosomal proteins BP180 and BP230 lead to dermoepidermal separation, causing tense blisters and/or pruritic plaques, predominantly in elderly individuals [4].

The epidemiology of AIBDs varies considerably across geographic regions and ethnic groups. The annual incidence of PV ranges from 0.6 to 32.0 cases per million, with a notable predisposition among individuals of Ashkenazi Jewish and Mediterranean descent [2,3]. In Turkey, the annual incidence of pemphigus, of which PV accounts for approximately 87.3%, was reported as 4.7 cases per million people [5]. The incidence rate of BP is generally higher, ranging from 18 to 76.3 cases per million person-years [4]. In Turkey, the incidence rate of pemphigoid diseases—of which BP is the predominant subtype, comprising 93.2% of cases—was reported as 3.55 cases per million person-years [6].

Growing evidence suggests that patients with pemphigus are at increased risk for comorbid autoimmune diseases (AIDs), including autoimmune thyroid diseases (AITDs), vitiligo, alopecia areata, rheumatoid arthritis (RA), type I diabetes mellitus (DM), systemic lupus erythematosus (SLE), and myasthenia gravis [7,8,9,10,11]. Similarly, associations between BP and psoriasis, multiple sclerosis, and AITDs have been reported [10,12,13]. However, population-based studies and meta-analyses have reported inconsistent associations, with some demonstrating variability in the associated diseases, whereas others found no association, highlighting population-level heterogeneity [14,15]. In a meta-analysis, an AID cluster including PV, AITD, RA, and type I DM, and another cluster including PV, SLE, AITD, and RA were proposed [8]. In a registry-based self-reported data of patients with BP, AITDs were the most common reported autoimmune comorbidities [16]. Similarly, Kridin et al. found an elevated burden of thyroiditis in patients with BP and PV in their population-based study [10].

Proposed mechanisms for the co-occurrence of AIDs, including shared genetic variants, environmental factors, cross-reactivity, human leukocyte antigen-mediated antigen presentation, shared signaling pathways, and generalized immune dysregulation, may also contribute to AIBD-associated autoimmunity; however, the direction and specificity of links remain unclear [8,17,18,19,20,21]. The recognition of such comorbidities is essential for comprehensive patient care, early diagnosis of coexisting disorders, and tailored management strategies.

Despite increasing recognition of autoimmune comorbidities in AIBDs, the spectrum of associated AIDs shows considerable variability across studies and populations. Heterogeneous study designs, varying diagnostic criteria, or reliance on self-reported data may complicate direct comparisons. Furthermore, the system-level patterns of autoimmune clustering in PV and BP have not been comprehensively characterized. Understanding these associations is essential to elucidate potential shared immunopathogenic pathways and to inform early screening and multidisciplinary management.

We hypothesized that PV and BP display higher rates of selective AIDs, not only those affecting the skin but also involving other organ systems, given the evidence of shared genetic loci, overlapping immunologic pathways, and previous reports of autoimmune clustering. We therefore aimed to assess the AIDS prevalence in PV and BP compared with matched controls and to describe the spectrum and system-level patterns of associated disorders.

## 2. Materials and Methods

### 2.1. Study Design

This bicenter, retrospective, cross-sectional, case–control study was conducted using electronic medical records from two tertiary referral centers in the Northern and Southern Marmara regions of Turkey. Both centers followed standardized diagnostic and data abstraction protocols, and diagnostic criteria for AIBDs and coexisting AIDs were identical across sites, based on established clinical, histopathologic, and serologic definitions. Data extraction was performed using predefined variables and a standardized electronic data collection template to ensure inter-center consistency and maintain data quality.

A retrospective search of records, based on International Classification of Diseases, Tenth Revision (ICD-10) codes (L10, L10.0, L10.8, L10.9, L12, L12.0, L12.8, and L12.9), was performed to identify PV and BP cases between January 2011 and December 2023. Diagnostic consistency between centers was ensured by standardized diagnostic criteria, including clinical evaluation, serologic testing (enzyme immunoassay/enzyme-linked immunosorbent assay detection of serum autoantibodies against desmoglein 3/1 for PV and BP180/BP230 when applicable), histopathologic examination, and direct immunofluorescence demonstrating intercellular or linear basement membrane IgG/C3 deposition for PV and BP, respectively. Patients were included if they had a confirmed diagnosis of PV or BP. Patients with uncertain diagnoses, missing clinical data, or other autoimmune bullous or dermatologic diseases were excluded.

Two separate control groups were recruited for the PV and BP groups. Using the electronic databases, we randomly selected four sex- and age-matched controls without any AIBD for each case from dermatology clinic attendees between 2011 and 2023. Controls were matched within the same center and time frame to minimize selection bias and ensure demographic comparability, with random selection from the electronic database to avoid investigator bias.

Medical records were reviewed for documented AIDS diagnoses, which were considered valid only if the patients were under regular follow-up and the diagnoses had been verified by the relevant specialty (e.g., endocrinology, rheumatology, neurology) according to current accepted clinical and serologic criteria. Cases with suspected or confirmed drug-induced autoimmune or rheumatic manifestations were excluded. In addition, thyroid autoimmunity was assessed by serum anti–thyroid peroxidase (anti-TPO) and anti-thyroglobulin (anti-Tg) levels, which were analyzed as serologic markers and did not by themselves define the diagnosis. Diagnosis of HT was accepted only when evaluated and confirmed by an endocrinologist, based on clinical or subclinical hypothyroidism with positive thyroid autoantibodies and/or characteristic ultrasonographic findings. Euthyroid HT cases confirmed by endocrinology consultation were also included. We collected age, sex, and detailed AIDS information and categorized AIDs by affected system per the literature [9,22].

### 2.2. Ethics Approval

This retrospective study was conducted using data obtained from medical records. The study was performed in accordance with the Declaration of Helsinki and was approved by the Non-Interventional Clinical Research Ethics Committee of Tekirdağ Namık Kemal University (protocol code: 2024.01.01.01; date of approval: 30 January 2024). According to the decision of the ethics committee, informed consent from individual patients was not required.

### 2.3. Statistical Analysis

Analyses were performed using SPSS v25.0 (IBM Corp., Armonk, NY, USA). Normality was assessed using the Kolmogorov–Smirnov test. Continuous variables were expressed as median (interquartile range [IQR]) and categorical variables as n (%). Comparisons between categorical variables were conducted using Pearson’s chi-square or Fisher’s exact test, as appropriate. Continuous variables were compared between two groups using the Mann–Whitney U test. Effect sizes were reported alongside *p*-values for all tests. Cramér’s V was used for chi-square analyses and Pearson’s r (derived from the Z-score) for the Mann–Whitney U test. Thresholds for small, medium, and large effects followed Cohen’s guidelines: 0.10, 0.30, and 0.50 for both Cramér’s V and r [23,24].

Multivariable logistic regression analysis (forward Wald method) was applied to evaluate independent risk factors for AIDs and the associations of PV and BP (independent variables) with binary AIDS outcomes. Variables with *p* < 0.20 in univariate analyses, where crude odds ratios (cORs) and 95% confidence intervals (CIs) were estimated, were included in the multivariable model to estimate adjusted odds ratios (aORs) with 95% CIs. Covariates included age (continuous), age range, sex, and concomitant additional AIDs. Model fit was assessed using the Hosmer–Lemeshow test. Statistical significance was set at *p* < 0.05.

## 3. Results

We included 287 patients with PV and 284 with BP, together with 1148 and 1137 sex- and age-matched controls. The PV and BP groups did not differ from their matched controls in sex, age (continuous), or age-range categories. Baseline demographics are provided in Table 1. Both groups showed a slight female predominance (female-to-male ratio, 1.5:1).

### 3.1. Association of AIBDs with AIDs

Overall, AID prevalence was significantly lower in PV (9.4%) and BP (7.7%) than in their respective controls (17.8% and 14.8%; Table 2). A complete list of AIDs by group is provided in Appendix A.

PV had a higher prevalence of Graves’ disease (GD) than matched controls (*p* = 0.036). By contrast, BP had a higher prevalence of Hashimoto thyroiditis (HT) but a lower prevalence of psoriasis than its controls (*p* = 0.014 and *p* = 0.002, respectively). No psoriasis cases were observed in the PV group (Table 2).

### 3.2. Distribution of AIDs by Affected System

Compared with controls, both PV and BP had significantly lower prevalence of cutaneous (*p* < 0.001 for each) and multisystem AIDs (*p* = 0.001 for each). Endocrine AIDs were more frequent in BP than in controls (*p* = 0.038), but not in PV. Gastrointestinal, neurological, hematological, or cardiovascular AIDs were rare and did not differ significantly between groups (Appendix A).

### 3.3. Stratified Analyses by Sex and Age

In both females and males, AIDS prevalence was lower in PV than in controls (*p* = 0.028 and *p* = 0.003, respectively). By contrast, in BP, AIDS prevalence was lower in females (*p* < 0.001), but not in males. Age-stratified analysis showed lower AIDS prevalence in the 41–60-year group with PV (*p* = 0.001) and in the 61–80-year group with BP (*p* = 0.005; Appendix A).

Among females, GD was more frequent in PV than in controls (*p* = 0.025), with no difference in males. Conversely, among males, HT was more frequent in BP than in matched controls (*p* = 0.011), with no difference in females. In the 61–80-year age group, GD was more prevalent in PV (*p* = 0.040); other age strata showed no differences. No age-stratum differences were observed for BP (Appendix A).

### 3.4. Logistic Regression Analyses

In multivariable models, PV was associated with decreased odds of any AID, whereas female sex was associated with increased odds (both moderate effect sizes). For GD, independent associations were observed for PV, female sex, and vitiligo; effect sizes were large for PV and female sex and very large for vitiligo, although the 95% CI for vitiligo was wide, indicating limited precision (Table 3).

In multivariable models, BP and age ≥ 81 years were associated with decreased odds of any AID, whereas female sex was associated with increased odds (all moderate effect sizes). For HT, independent associations were observed for BP, alopecia areata, and vitiligo, with a moderate effect size for BP and very large effect sizes for alopecia areata and vitiligo. Notably, the CIs for alopecia areata and vitiligo were wide, indicating limited precision (Table 3).

In affected-system analyses, PV was independently associated with decreased odds of cutaneous and multiple AIDs, both with large effect sizes. Endocrine, gastrointestinal, and multisystem AIDs showed independent associations with cutaneous AIDs (all large effect sizes), though the estimate for gastrointestinal AIDs was imprecise owing to a wide 95% CI. For multisystem AIDs, independent associations beyond PV were observed for female sex and cutaneous AIDs, with moderate and large effect sizes, respectively (Table 4).

BP was associated with decreased odds of cutaneous and multisystem AIDs (both large effect sizes). Endocrine and multisystem AIDs were associated with increased odds of cutaneous AIDs (both large effect sizes). For multisystem AIDs, female sex and cutaneous AIDs were independently associated with increased odds. BP and cutaneous AIDs were associated with increased odds of endocrine AIDs, with moderate and large effect sizes, respectively (Table 5).

### 3.5. Comparisons of Laboratory Tests

Comparisons of thyroid function tests and thyroid autoantibodies, including sex- and age-stratified analyses, are shown in Appendix A. Overall anti-TPO and anti-Tg positivity did not differ significantly between BP and controls. In PV, anti-Tg positivity was less frequent than in controls (*p* = 0.017), whereas anti-TPO positivity did not differ (*p* = 0.881).

## 4. Discussion

In this bicenter, case–control study of 571 patients with AIBDs (287 PV; 284 BP), overall AID prevalence was lower in both PV and BP than in matched controls. Nevertheless, disease-specific associations emerged: GD was more frequent in PV, particularly among females, while HT was more common in BP and most evident among males compared to controls. Multivariable models corroborated these findings: PV and BP were independently associated with reduced odds of any AID, whereas PV had higher odds of GD and BP had higher odds of HT. Vitiligo (PV model) and vitiligo/alopecia areata (BP model) carried very large effect sizes for thyroid autoimmunity; however, with notably wide CIs, indicating limited precision.

Beyond individual diseases, system-level analyses showed that cutaneous and multisystem AIDs were markedly less prevalent in PV and BP than in controls, while endocrine AIDs were enriched in BP. These trends remained after adjustment: both PV and BP appeared to be independently associated with lower odds of cutaneous/multisystem AIDs, whereas BP may be independently associated with higher odds of endocrine AIDs.

### 4.1. Association of AIBDs with AIDs

Prior studies have reported associations between PV/BP and various AIDs [7,8,9,10,12,13,20,21]. A meta-analysis indicated that PV belongs to two distinct AID clusters and shows a higher prevalence of AITD, RA, and type I DM than in the general population, implying shared etiopathogenesis [8]. Narla et al., in their nationwide U.S. study, reported increased odds of having at least one AID among inpatients with pemphigus or pemphigoid, as well as higher odds of multiple cutaneous and systemic AIDs. The strongest associations were observed between unspecified AID, vitiligo, and eosinophilic esophagitis in pemphigus inpatients, and between unspecified AID, vitiligo, and chronic urticaria in pemphigoid inpatients [7]. In a retrospective case–control study from Italy, dermatologic autoimmune conditions, including psoriasis, lichen planus, vitiligo, alopecia areata, and lichen sclerosus, were independently associated with pemphigus vulgaris [25]. In addition, a large-scale global cohort study found that both pemphigus and BP were associated with cutaneous and systemic lupus erythematosus, alopecia areata, and connective tissue diseases such as dermatomyositis and Sjögren’s disease [26]. Saurabh et al. reported an increased likelihood of developing pemphigoid diseases, discoid lupus erythematosus, lichen planus, and undifferentiated connective tissue disease among pemphigus patients across all ethnic groups. However, certain AIDs, such as antiphospholipid syndrome, psoriasis, and myasthenia gravis, exhibited ethnic variability [9]. The discrepancies in previous findings may, in part, reflect differences in study region, methodology, and ethnic background of populations investigated.

Our analysis demonstrated a lower overall AID prevalence in both PV and BP compared with matched controls. Both diseases were associated with lower odds of any AID, likely reflecting the high prevalence of cutaneous AIDs among dermatology clinic-based controls. Nevertheless, disease-specific analyses yielded robust and biologically plausible associations: PV was independently linked to GD and BP to HT, suggesting a more targeted, phenotype-driven pattern of autoimmunity rather than a broad autoimmune diathesis. This pattern suggests that focusing on specific, biologically plausible associations, rather than aggregate measures of any AID, may better guide screening approaches and management strategies.

By affected systems, PV and BP were independently associated with lower odds of cutaneous and multisystem AIDs, whereas BP showed higher odds of endocrine AIDs. Beyond AIBDs themselves, regression analyses suggested reciprocal associations among cutaneous, endocrine, and multisystem AIDs. Together, these findings point to distinct system-level comorbidity profiles rather than a uniform autoimmune predisposition and raise the hypothesis of shared pathogenic pathways that could transcend organ boundaries.

### 4.2. Distinct Thyroid Autoimmunity and AIBD Profiles

Associations between AITDs and pemphigus or BP have largely been reported, although effect sizes and directionality vary across populations and study designs [8,9,10,11,16,27]. However, these associations were not supported in a population-based study [15]. In a nationwide study, HT and thyroiditis were associated with pemphigus and BP only in persons < 50 years of age. Moreover, HT was only associated with pemphigus in female patients [7]. A large-scale cross-sectional study consisting of patients with pemphigus indicated a significant association with HT only in males but not in all patients, while no association was found between pemphigus and GD [28]. In our study, the selective enrichment of GD in PV, particularly among females, and HT in BP, particularly among males, suggested phenotype-specific thyroid autoimmunity rather than a generalized increase across all AIDs. Co-occurrence of PV and GD may suggest that sex-related immune or hormonal factors modulate susceptibility to concomitant autoimmunity. However, although thyroid autoimmunity typically exhibits a marked female predominance, HT appeared to be particularly relevant among men with BP. Notably, HT prevalence was also numerically higher among women with BP than in controls, although this difference did not reach statistical significance, indicating a consistent but more pronounced trend in males. This observation may reflect distinct immunogenetic mechanisms in BP or sex-specific differences in immune regulation that modify thyroid autoimmunity risk once pemphigoid is established. The relatively low HT prevalence in BP controls likely reflects the clinic-based sample and the use of strict, specialist-confirmed diagnoses rather than serology-based or self-reported definitions.

The association between cutaneous autoimmunity, particularly vitiligo and alopecia areata, and thyroid disease has been consistently observed in other populations [29,30,31]. In our study, the very large effect sizes for vitiligo in the PV model and for both vitiligo and alopecia areata in the BP model with respect to thyroid autoimmunity were particularly striking. However, the CIs for these associations were very wide, most likely due to the small number of cases, indicating limited precision. Nevertheless, given the well-established links between vitiligo, alopecia areata, and thyroid autoimmunity, it is plausible that their coexistence in patients with PV or BP is associated with a higher likelihood of concomitant AITD. Future studies in larger, prospective, and ethnically diverse cohorts are warranted to confirm this hypothesis.

On the other hand, anti-TPO/Tg positivity did not parallel the clinical associations. Anti-TPO antibody frequencies did not differ between PV or BP patients and their controls, whereas anti-Tg antibody positivity was significantly lower in the PV group compared with controls. Consistent with this, one clinical series reported no significant differences in anti-TPO and anti-Tg antibody levels between the patients with PV and healthy controls [32]. In contrast, several small controlled studies have reported higher anti-TPO positivity in both PV and BP [11,27,33,34]. However, in two small case–control studies, despite more frequent positive anti-TPO antibody titers in PV, no association with overt clinical AITD was observed [33,34]. By contrast, Kavala et al., in their case–control study from Turkey, reported that positive anti-TPO antibody and HT were more frequent in the PV group, whereas the frequency of anti-Tg antibody positivity showed no significant difference [35]. The frequent use of systemic immunosuppressive agents such as corticosteroids in AIBDs may contribute to reduced autoantibody production and could partly explain these discrepancies. Moreover, the reliance on single-point antibody measurements and the distinction between serologic and clinical disease may represent additional factors. Taken together, these findings suggest that routine screening for thyroid autoantibodies may not be warranted in all AIBD patients. Instead, an immunologic or genetic subset may be predisposed to thyroid autoimmunity, even in the presence of low serum autoantibody levels. Further prospective studies are required to identify this susceptible subgroup and to determine whether targeted screening strategies could provide clinical benefit.

### 4.3. AIBDs and Psoriasis

An association between BP and psoriasis has been reported [12,13,20,36,37,38]. In a population-based study, Kridin et al. demonstrated a bidirectional causal association [12], whereas Zhang et al. found psoriasis vulgaris increased the risk of BP but not vice versa, suggesting a unidirectional relationship [13]. A meta-analysis demonstrated that psoriasis was significantly more prevalent among patients with BP than among controls [38]. However, the prevalence of psoriasis among patients with BP varies across populations, suggesting a potential ethnic predisposition to this association [39]. Data on the exact prevalence of psoriasis among BP patients in Turkey are currently lacking. The inverse association we observed between BP and psoriasis contrasts with these large epidemiologic analyses. Control selection likely contributed, as our controls were recruited from dermatology clinics where psoriasis may be relatively common, thereby elevating the baseline prevalence of cutaneous AIDs. Retrospective coding could also have led to underdetection of concomitant psoriasis. In addition, the divergent immunologic profiles of BP, which is generally characterized by a Th2/eosinophilic response, and psoriasis, which is driven by Th17/Th1 pathways, may biologically reduce co-occurrence in certain populations. Finally, population differences in genetics, environmental exposures, and therapeutic interventions may further contribute to the heterogeneity observed across studies [9,40,41].

The association between psoriasis and various forms of pemphigus has been reported in case reports and case series. In a Japanese case series of patients with concomitant psoriasis and AIBDs, pemphigus foliaceus was identified in 2.8% of cases [36]. Population-based cross-sectional studies have further supported this link, demonstrating a significant association between psoriasis and pemphigus [42,43,44]. Meta-analyses have also revealed a significant relationship [45,46], although the exact link remains unclear [21]. In contrast, we observed no cases of concomitant psoriasis and PV in our cohort, underscoring the need to investigate population-specific, methodological, and immunological factors that may explain the variability across studies.

### 4.4. Interpretation of the Findings

These findings suggest that selective, phenotype-specific screening rather than blanket testing for all AIDs may be more appropriate in AIBDs. In PV, greater emphasis on GD, particularly among females, may be warranted, whereas in BP, HT could be prioritized, especially in males, with a low threshold for thyroid function testing when vitiligo or alopecia areata co-occur.

The relative depletion of cutaneous AIDs and enrichment of endocrine AIDs in BP suggest that a targeted medical history, review of systems, and laboratory workup could be beneficial. However, the clinical relevance of these findings should be interpreted with caution, as effect sizes were generally small, and larger studies are needed to determine whether these modest associations reflect true underlying patterns. Moreover, causal relationships cannot be inferred due to the lack of temporal data. Finally, ancestry-driven heterogeneities in autoimmune clustering are well recognized [40,41,47], and prospective studies with replication in independent and ethnically diverse cohorts are required to validate our results.

### 4.5. Future Directions

Prospective cohorts with standardized AIDS screening at and after AIBD diagnoses are needed to establish temporality and absolute risks, incorporate drug exposures and immune endophenotypes, and validate sex- and age-specific effects. Integrative multi-omics with fine-mapping at shared susceptibility loci could clarify whether the PV–GD and BP–HT signals reflect pleiotropy, tissue microenvironment, or systemic immune programming, and may yield biomarkers to personalize surveillance and therapy [13,31,32].

### 4.6. Strengths and Limitations of the Study

Strengths of this study include rigorous diagnostic confirmation (clinical, histopathology, direct immunoflorescence, and serologic), age- and sex-matched controls, and parallel disease-specific and system-level analyses with multivariable modeling. The comprehensive evaluation of autoimmune comorbidities across organ systems, rarely addressed in previous literature, enabled the identification of previously unreported patterns—lower odds of cutaneous and multisystem AIDs in both PV and BP, and higher odds of endocrine AIDs in BP. These findings provide novel insights into AIBD-associated autoimmunity and its population-specific variability.

This study has several limitations. The retrospective design may have led to under-ascertainment of coexisting AIDs. Because the temporality of diagnoses could not be established, causal inference is precluded. The use of dermatology clinic-based controls may have inflated the baseline prevalence of cutaneous AIDs, introducing control group bias. Importantly, key potential confounders such as smoking, alcohol use, immunosuppressive medication, socioeconomic status, and comorbidities were not available and therefore could not be adjusted for. All patients with AIBDs were treated with systemic corticosteroids and/or other immunosuppressive or immunomodulatory agents (e.g., azathioprine, mycophenolate mofetil, rituximab); however, treatment data were not available for the control group, precluding comparative analyses by immunosuppressive exposure. This unmeasured confounding, along with limited power for rarer AIDs and certain sex- and age-specific strata, and restricted generalizability beyond Turkish tertiary-care settings, further limits interpretation. In addition, extended autoantibody data (e.g., rheumatoid factor, anti-cyclic citrullinated peptide, extractable nuclear antigen panel) were not systematically available for the entire cohort, except for certain specific AIDs such as rheumatologic diseases, and therefore could not be analyzed. Only ANA results were available for a subset of patients and were reported descriptively. Consequently, disease-specific autoantibody comparisons were not feasible; however, diagnostic accuracy was ensured, as all AID diagnoses were specialist-confirmed based on clinical and serologic criteria. Many autoimmune comorbidities were analyzed without applying formal corrections for multiple comparisons, which may have increased the likelihood of false-positive results. However, the use of multivariable logistic regression helped to account for potential confounders and reduce random associations; therefore, the findings should be interpreted with caution and regarded as exploratory. Because autoimmune clustering varies by ancestry, validation in independent, ethnically diverse, population-based cohorts is needed to validate these exploratory findings.

## 5. Conclusions

Our findings suggest that AIBDs may display selective, rather than generalized, autoimmune comorbidity patterns. Although both PV and BP showed lower overall odds of AIDs compared with matched controls, disease-specific trends were observed: PV appeared to be associated with GD, particularly among females, whereas BP showed a possible association with HT, especially among males. BP also appeared to have higher odds of endocrine AIDs overall, while both PV and BP exhibited lower odds of cutaneous and multisystem AIDs. Interestingly, in contrast to previous reports, psoriasis prevalence was lower in both PV (none) and BP (0.7%) compared with controls. However, effect sizes were generally small across analyses, indicating limited clinical relevance. Moreover, thyroid autoantibody positivity rates did not parallel the clinical associations, suggesting limited screening utility in AIBDs. Taken together, these exploratory and hypothesis-generating findings may reflect sex- and system-specific autoimmune clustering, warranting confirmation in larger, prospective, and ethnically diverse cohorts to clarify temporality and shared immunopathogenic mechanisms before informing routine clinical practice.

## Figures and Tables

**Table 1 medicina-61-01946-t001:** Demographics of study groups.

Variables	PV Group *n* = 287	Control Group *n* = 1148	*p*-Value
**Sex (*n*, %)**			>0.999
Female	172 (59.9%)	688 (59.9%)	
Male	115 (40.1%)	460 (40.1%)	
Female:male	1.5:1	1.5:1	
**Age (yrs) (median, IQR)**	57 (46–66)	55 (45–66)	0.327
**Age range (yrs) (*n*, %)**			>0.999
<20	3 (1%)	12 (1%)	
21–40	34 (11.8%)	133 (11.6%)	
41–60	143 (49.8%)	577 (50.3%)	
61–80	91 (31.7%)	364 (31.7%)	
≥81	16 (5.6%)	62 (5.4%)	
**Variables**	**BP Group** ** *n* ** **= 284**	**Control Group** ** *n* ** **= 1137**	** *p* ** **-Value**
**Sex (*n*, %)**			0.987
Female	170 (59.9%)	680 (59.8%)	
Male	114 (40.1%)	457 (40.2%)	
Female:male	1.5:1	1.5:1	
**Age (yrs) (median, IQR)**	74 (67–80)	72 (66–79)	0.077
**Age range (yrs) (*n*, %)**			0.977
41–60	31 (10.9%)	129 (11.3%)	
61–80	191 (67.3%)	759 (66.8%)	
≥81	62 (21.8%)	249 (21.9%)	

PV, pemphigus vulgaris; n, number; yrs, years; IQR, interquartile range; BP, bullous pemphigoid.

**Table 2 medicina-61-01946-t002:** Comparison of the frequencies of overall AIDs, HT, GD, and psoriasis between cases and controls.

AIBD	*n* (%) of Overall AID in Cases	*n* (%) of Overall AID in Controls	cOR (95% CI)	*p*-Value	Effect Size *
**PV**	27 (9.4%)	204 (17.8%)	0.481 (0.314–0.734)	**0.001**	0.091
**BP**	22 (7.7%)	168 (14.8%)	0.484 (0.304–0.771)	**0.002**	0.083
**AIBD**	** *n* ** **(%) of HT in Cases**	** *n* ** **(%) of HT in Controls**	**cOR (95% CI)**	** *p* ** **-Value**	**Effect Size ***
PV	15 (5.2%)	56 (4.9%)	1.075 (0.599–1.930)	0.808	–
BP	16 (5.6%)	31 (2.7%)	2.128 (1.147–3.948)	**0.014**	0.065
**AIBD**	** *n* ** **(%) of GD in Cases**	** *n* ** **(%) of GD in Controls**	**cOR (95% CI)**	** *p* ** **-Value**	**Effect Size ***
PV	8 (2.8%)	11 (1%)	2.964 (1.181–7.438)	**0.036**	0.064
BP	3 (1.1%)	4 (0.4%)	3.021 (0.672–13.577)	0.148	–
**AIBD**	** *n* ** **(%) of Psoriasis in Cases**	** *n* ** **(%) of Psoriasis in Controls**	**cOR (95% CI)**	** *p* ** **-Value**	**Effect Size ***
PV	0	48 (4.2%)	–	**<0.001**	0.093
BP	2 (0.7%)	52 (4.6%)	0.148 (0.036–0.611)	**0.002**	0.081

AIBD, autoimmune bullous disease; n, number; AID, autoimmune disease; cOR, crude odds ratio; CI, confidence interval; PV, pemphigus vulgaris; BP, bullous pemphigoid; HT, Hashimoto thyroiditis; GD, Graves’ disease. * Cramér’s V. Statistically significant values (*p* < 0.05) are highlighted in bold.

**Table 3 medicina-61-01946-t003:** Logistic regression analysis of risk factors for AIDs (overall, GD, and HT) in PV, BP and their matched controls.

Cohort	Predictors	Outcome	Univariate			Multivariate		
			cOR	95% CI	*p*-Value	aOR	95% CI	*p*-Value
PV and matched controls	Pemphigus	AID	0.481	0.314–0.734	**0.001**	0.477	0.312–0.730	**0.001**
	Sex (Female)		1.877	1.378–2.556	**<0.001**	1.886	1.384–2.571	**<0.001**
	Older age		0.994	0.985–1.004	0.232	–	–	–
**Cohort**	**Predictors**	**Outcome**	**Univariate**			**Multivariate**		
			**cOR**	**95% CI**	** *p* ** **-Value**	**aOR**	**95% CI**	** *p* ** **-Value**
PV and matched controls	Pemphigus	GD	2.964	1.181–7.438	**0.021**	3.155	1.235–8.058	**0.016**
	Sex (Female)		3.845	1.122–13.180	**0.032**	3.585	1.030–12.481	**0.045**
	Older age		1.009	0.979–1.040	0.544	–	–	–
	Vitiligo		18.392	3.695–91.560	**<0.001**	18.979	3.628–99.283	**<0.001**
	RA		6.500	0.802–52.673	0.080	–	–	–
**Cohort**	**Predictors**	**Outcome**	**Univariate**			**Multivariate**		
			**cOR**	**95% CI**	** *p* ** **-Value**	**aOR**	**95% CI**	** *p* ** **-Value**
BP and matched controls	BP	AID	0.484	0.304–0.771	**0.002**	0.479	0.300–0.765	**0.002**
	Sex (Female)		1.878	1.341–2.630	**<0.001**	1.870	1.333–2.624	**<0.001**
	Age range (≥81 years)		0.657	0.499–0.865	**0.003**	0.662	0.503–0.873	**0.003**
**Cohort**	**Predictors**	**Outcome**	**Univariate**			**Multivariate**		
			**cOR**	**95% CI**	** *p* ** **-Value**	**aOR**	**95% CI**	** *p* ** **-Value**
BP and matched controls	BP	HT	2.128	1.147–3.948	**0.017**	2.509	1.326–4.749	**0.005**
	Sex (Female)		1.604	0.851–3.025	0.144	–	–	–
	Age range		1.007	0.248–4.080	0.993	–	–	–
	AA		11.634	2.985–45.346	**<0.001**	16.721	4.154–67.300	**<0.001**
	Vitiligo		11.634	2.985–45.346	**<0.001**	14.089	3.542–56.038	**<0.001**

AID, autoimmune disease; cOR, crude odds ratio; CI, confidence interval; aOR, adjusted odds ratio; GD, Graves’ disease; RA, rheumatoid arthritis; BP, bullous pemphigoid; HT, Hashimoto thyroiditis; AA, alopecia areata. Statistically significant values (*p* < 0.05) are highlighted in bold.

**Table 4 medicina-61-01946-t004:** Logistic regression analysis of risk factors for AIDs, affecting cutaneous or multiple systems, in the cohort of PV and matched controls.

Cutaneous AIDs	Univariate			Multivariate		
	cOR	95% CI	*p*-Value	aOR	95% CI	*p*-Value
PV	0.192	0.077–0.476	**<0.001**	0.211	0.084–0.527	**0.001**
Older age	0.976	0.963–0.990	**<0.001**	0.974	0.960–0.988	**<0.001**
Endocrine AIDs	2.760	1.467–5.191	**0.002**	3.117	1.607–6.047	**0.001**
GIS AIDs	13.310	1.855–95.487	**0.010**	9.259	1.143–75.016	**0.037**
Multisystem AIDs	4.773	2.515–9.057	**<0.001**	4.260	2.176–8.340	**<0.001**
**Multisystem AIDs**	**Univariate**			**Multivariate**		
	**cOR**	**95% CI**	** *p* ** **-Value**	**aOR**	**95% CI**	** *p* ** **-Value**
PV	0.139	0.034–0.575	**0.006**	0.163	0.039–0.676	**0.012**
Sex (Female)	2.109	1.143–3.892	**0.017**	2.186	1.176–4.062	**0.013**
Older age	1.012	0.994–1.030	0.204			
Cutaneous AIDs	4.773	2.515–9.057	**<0.001**	4.219	2.205–8.074	**<0.001**
Endocrine AIDs	1.671	0.648–4.305	0.288			
GIS AIDs	8.185	0.838–79.93	0.071			

AID, autoimmune disease; cOR, crude odds ratio; CI, confidence interval; aOR, adjusted odds ratio; PV, pemphigus vulgaris; GIS, gastrointestinal system. Statistically significant values (*p* < 0.05) are highlighted in bold.

**Table 5 medicina-61-01946-t005:** Logistic regression analysis of risk factors for AIDs, affecting cutaneous, endocrine, or multiple systems, in the cohort of BP and matched controls.

Cutaneous AIDs	Univariate			Multivariate		
	cOR	95% CI	*p*-Value	aOR	95% CI	*p*-Value
BP	0.143	0.045–0.456	**0.001**	0.141	0.044–0.457	**0.001**
Age group (≥81 years)	0.099	0.028–0.349	**<0.001**	0.103	0.029–0.367	**<0.001**
Endocrine AIDs	4.397	2.117–9.132	**<0.001**	5.294	2.418–11.590	**<0.001**
GIS AIDs	5.498	0.566–53.444	0.142			
Multisystem AIDs	5.294	2.668–10.503	**<0.001**	3.984	1.964–8.080	**<0.001**
**Endocrine AIDs**	**Univariate**			**Multivariate**		
	**cOR**	**95% CI**	** *p* ** **-Value**	**aOR**	**95% CI**	** *p* ** **-Value**
BP	1.880	1.025–3.447	**0.041**	2.311	1.231–4.340	**0.009**
Sex (Female)	1.639	0.889–3.023	0.113			
Neurologic AIDs	4.547	0.537–38.479	0.165			
Cutaneous AIDs	4.397	2.117–9.132	**<0.001**	5.354	2.511–11.415	**<0.001**
**Multisystem AIDs**	**Univariate**			**Multivariate**		
	**cOR**	**95% CI**	** *p* ** **-Value**	**aOR**	**95% CI**	** *p* ** **-Value**
BP	0.072	0.010–0.525	**0.009**	0.085	0.012–0.623	**0.015**
Sex (Female)	3.063	1.528–6.137	**0.002**	3.067	1.522–6.181	**0.002**
Age group (≥81 years)	0.348	0.109–1.114	0.075			
Cutaneous AIDs	5.294	2.668–10.503	**<0.001**	4.380	2.182–8.792	**<0.001**

AID, autoimmune disease; cOR, crude odds ratio; CI, confidence interval; aOR, adjusted odds ratio; BP, bullous pemphigoid; GIS, gastrointestinal system. Statistically significant values (*p* < 0.05) are highlighted in bold.

## Data Availability

The original contributions presented in this study are included in the article/Appendix A. Further inquiries can be directed to the corresponding author.

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
