# Peer review of "Sex-Specific Autoimmune Comorbidity Patterns in Pemphigus Vulgaris and Bullous Pemphigoid: A Bicenter Retrospective Case–Control Study"

_medicina, 2025, doi:10.3390/medicina61111946_

Round 1
Reviewer 1 Report
Comments and Suggestions for Authors
- Include some Pemphigus Vulgaris and Bullous Pemphigoid statistics in the introduction part.
- Add explanation regarding Pemphigus Vulgaris and Bullous Pemphigoid in the introduction part along with research gap or problem statement and study rationale.
- please explain what were the inclusion and exclusion criteria for selecting the patients with pemphigus vulgaris (PV) and bullous pemphigoid (BP), and how were controls matched to minimize selection bias during the data collection?
- The author mentioned that this was a bicenter retrospective study, how were differences in diagnostic criteria and data collection standardized between the two centers? why this study was not conducted for multicenter to achieved better and more data?
- Is mentioned in the title of article that is Phenotype- and Sex-Specific Autoimmune Comorbidity Patterns so how were the phenotypes of PV and BP defined and classified and what were the standardized clinical criteria example i.e., ELISA or any other diagnostics tools for autoantibodies which were used to ensure consistency?
- Is mentioned in the title regarding the Sex-Specific Autoimmune Comorbidity Patterns so what was the sex distribution within each group and what was the study adequately properly powered to detect sex-related difference Comorbidity Patterns? please explain
- how did the authors handle multiple comparisons to avoid false-positive finding between automines comorbidities, phenotype and sex? please explain the logic behind it.
- the authors were point out or studied any novel type of associations that have not been reported in previous published literature? please explain if you have
- the authors are requested to enrich the discussion section with the previous published literature
- the authors are requested to rewrite the conclusion part supported by the results
Author Response
Thank you for your valuable comments. All comments were responsed and all revisions were made point-by-point accordingly. Thank you for your consideration.
The detailed responses have also been submitted in Word format.
REVIEWER 1
1.Include some Pemphigus Vulgaris and Bullous Pemphigoid statistics in the introduction part.
In the Introduction section, we have added epidemiological data on pemphigus vulgaris and bullous pemphigoid.
2. Add explanation regarding Pemphigus Vulgaris and Bullous Pemphigoid in the introduction part along with research gap or problem statement and study rationale.
We have added brief explanations of pemphigus vulgaris and bullous pemphigoid in the introduction to provide background on their clinical and immunopathological characteristics. In addition, we have clarified the research gap and the rationale for the present study.
3. Please explain what were the inclusion and exclusion criteria for selecting the patients with pemphigus vulgaris (PV) and bullous pemphigoid (BP), and how were controls matched to minimize selection bias during the data collection?
We have now clearly defined the inclusion and exclusion criteria for patients with pemphigus vulgaris and bullous pemphigoid in the “Study Design/Methods” section:
“A retrospective search of records, based on International Classification of Diseas-es, Tenth Revision (ICD-10) codes (L10, L10.0, L10.8, L10.9, L12, L12.0, L12.8, and L12.9), was performed to identify PV and BP cases between January 2011 and Decem-ber 2023. Diagnostic consistency between centers was ensured by standardized diag-nostic criteria, including clinical evaluation, serologic testing (enzyme immunoas-say/enzyme-linked immunosorbent assay detection of serum autoantibodies against desmoglein 3/1 for PV and BP180/BP230 when applicable), histopathologic examina-tion, and direct immunofluorescence demonstrating intercellular or linear basement membrane IgG/C3 deposition for PV and BP, respectively. Patients were included if they had a confirmed diagnosis of PV or BP. Patients with uncertain diagnoses, missing clinical data, or other autoimmune bullous or dermatologic diseases were excluded.”
We have also elaborated on the process of control selection and matching to clarify how potential selection bias was minimized during data collection: “Controls were matched within the same center and time frame to minimize selection bias and ensure demographic comparability, with random selection from the electronic database to avoid investigator bias.”
4. The author mentioned that this was a bicenter retrospective study, how were differences in diagnostic criteria and data collection standardized between the two centers? why this study was not conducted for multicenter to achieved better and more data?
Although this was a retrospective study, both centers are tertiary dermatology referral hospitals located in Marmara Region of Turkey with comparable patient populations and follow standardized diagnostic and treatment protocols for autoimmune bullous diseases. Diagnostic criteria for pemphigus vulgaris, bullous pemphigoid, and associated autoimmune diseases were identical across centers and based on established clinical, histopathologic, and serologic definitions. Data extraction was performed using predefined variables and a harmonized electronic data form to ensure consistency.
We intentionally limited the study to two centers to maintain diagnostic uniformity and data quality. A larger multicenter design could indeed increase generalizability; however, retrospective variability in diagnostic documentation across different institutions might introduce bias. The bicenter design allowed for a sufficiently large and well-characterized sample while minimizing inter-center variability. Future prospective multicenter studies with unified data collection frameworks are warranted to validate and expand upon our findings.
“Both centers followed standardized diagnostic and data abstraction protocols, and diagnostic criteria for AIBDs and coexisting AIDs were identical across sites, based on established clinical, histopathologic, and serologic definitions. Data extraction was performed using predefined variables and a standardized electronic data collection template to ensure inter-center consistency and maintain data quality.” These explanations were added to Methods/Study Design section.
5. Is mentioned in the title of article that is Phenotype- and Sex-Specific Autoimmune Comorbidity Patterns so how were the phenotypes of PV and BP defined and classified and what were the standardized clinical criteria example i.e., ELISA or any other diagnostics tools for autoantibodies which were used to ensure consistency?
We appreciate the reviewer’s comment and agree that clarification was needed. In our study, “phenotype” refers to the disease type (PV or BP) as distinct clinical entities within autoimmune bullous diseases, each defined by characteristic clinical, histopathologic, and immunologic features. However, we did not perform a separate clinical or serologic phenotype classification within the PV and BP groups. The term “phenotype-specific” in the original title may therefore have caused confusion, and we have removed it to better reflect the true scope of the study.
In both participating centers, standardized diagnostic criteria were applied to ensure consistency. In the Methods section, revisions were made accordingly: “Diagnostic consistency between centers was ensured by standardized diagnostic criteria, including clinical evaluation, serologic testing (enzyme immunoas-say/enzyme-linked immunosorbent assay detection of serum autoantibodies against desmoglein 3/1 for PV and BP180/BP230 when applicable), histopathologic examination, and direct immunofluorescence demonstrating intercellular or linear basement membrane IgG/C3 deposition for PV and BP, respectively.”
6. Is mentioned in the title regarding the Sex-Specific Autoimmune Comorbidity Patterns so what was the sex distribution within each group and what was the study adequately properly powered to detect sex-related difference Comorbidity Patterns? please explain
The sex distribution within the study groups was as follows: in the PV group, 172 females (59.9%) and 115 males (40.1%); in the BP group, 170 females (59.9%) and 114 males (40.1%); in the PV-control group, 688 females (59.9%) and 460 males (40.1%); in the BP-control group, 680 females (59.8%) and 457 males (40.2%).
As this was a retrospective, bicenter case–control study based on existing medical records, no a priori power analysis was performed. However, to enhance statistical power and reduce sampling bias, four age- and sex-matched controls were randomly selected for each case (1:4 case-to-control ratio). Given the relatively large sample size and 1:4 matching design, with data collected from two tertiary centers over a 12-year period, the study was adequately powered for descriptive assessment of sex-related differences in autoimmune comorbidity patterns. The findings are therefore intended to be exploratory and hypothesis-generating rather than confirmatory.
7. how did the authors handle multiple comparisons to avoid false-positive finding between automines comorbidities, phenotype and sex? please explain the logic behind it.
As multiple autoimmune comorbidities and sex-related comparisons were analyzed, we acknowledge the potential risk of false-positive findings due to multiple testing. Because the study was exploratory and descriptive in nature, no formal correction for multiple comparisons was applied. Instead, multivariable logistic regression analyses were performed to reduce random associations and control for potential confounders. This approach provided adjusted estimates and improved the robustness of the results.
We have also added a statement in the Limitations section to acknowledge this issue as follows:
“Many autoimmune comorbidities were analyzed without applying formal corrections for multiple comparisons, which may have increased the likelihood of false-positive results. However, the use of multivariable logistic regression helped to account for potential confounders and reduce random associations; therefore, the findings should be interpreted with caution and regarded as exploratory.”
8. the authors were point out or studied any novel type of associations that have not been reported in previous published literature? please explain if you have
The primary aim of our study was not to investigate specific autoimmune disease pairs but to comprehensively assess the overall spectrum and system-level patterns of autoimmune comorbidities in patients with PV and BP using standardized clinical and serologic criteria. To our knowledge, this organ- or system-based analytical approach has not been previously reported in the literature and represents a methodological novelty of our work. In addition, diagnoses were not based solely on ICD codes or self-reported data. Medical records were reviewed in detail, and only cases with confirmed diagnoses established through clinical, histopathologic, serologic, and direct immunofluorescence findings were included.
Regarding our findings, both PV and BP were independently associated with lower odds of cutaneous and multisystem autoimmune diseases, while BP demonstrated higher odds of endocrine autoimmune diseases—patterns not previously described in population-based studies.
In contrast to earlier reports suggesting a positive association between AIBDs and psoriasis, our study did not find such an association; notably, no psoriasis cases were observed in the PV group, and only two cases (0.7%) were identified in the BP group. These results highlight possible population-specific variations in autoimmune clustering and may provide new insights into the heterogeneity of AIBD-associated autoimmunity. Further studies in independent and ethnically diverse cohorts are warranted to confirm these novel observations.
“The comprehensive evaluation of autoimmune comorbidities across organ systems, rarely addressed in previous literature, enabled the identification of previously unreported patterns—lower odds of cutaneous and multisystem AIDs in both PV and BP, and higher odds of endocrine AIDs in BP. These findings provide novel insights into AIBD-associated autoimmunity and its population-specific variability.” This part was added to “Strengths and Limitations of the Study”.
9. the authors are requested to enrich the discussion section with the previous published literature
The Discussion section has been enriched with additional references and a more detailed comparison to previously published literature to provide broader context and support for our findings.
10. the authors are requested to rewrite the conclusion part supported by the results
We have rewritten the conclusion section to better reflect and support the study findings, as suggested:
“Our findings suggest that AIBDs may display selective, rather than generalized, autoimmune comorbidity patterns. Although both PV and BP showed lower overall odds of AIDs compared with matched controls, disease-specific trends were observed: PV was associated with GD, particularly among females, whereas BP was associated with HT, especially among males. BP also showed higher odds of endocrine AIDs overall, while both PV and BP exhibited lower odds of cutaneous and multisystem AIDs. Interestingly, in contrast to previous reports, psoriasis prevalence was lower in both PV (none) and BP (0.7%) compared with controls. However, effect sizes were generally small across analyses, indicating limited clinical relevance. Moreover, thyroid autoantibody positivity rates did not parallel the clinical associations, suggesting limited screening utility in AIBDs. Taken together, these exploratory and hypothesis-generating findings may reflect sex- and system-specific autoimmune clustering, warranting confirmation in larger, prospective, and ethnically diverse cohorts to clarify temporality and shared immunopathogenic mechanisms before informing routine clinical practice.”

Reviewer 2 Report
Comments and Suggestions for Authors
The manuscript addresses a relevant topic. The study is well organized, with clear methodology, solid statistical analyses.
The retrospective design significantly limits the strength of causal inferences, yet in several places, the conclusions are presented too strongly. The authors should moderate their statements and clarify that the results reflect associations. Important confounding factors such as smoking, alcohol use, immunosuppressive medication, and social and economic status were not considered, their absence should be explicitly acknowledged as a limitation.
It would be helpful to clarify whether the diagnosis of associated autoimmune diseases was verified by specialists or based solely on medical records of patient s pathological history.
Tables are clear, but highlighting statistically significant associations would improve readability. The abstract and conclusion are currently written in a relatively strong tone and should be revised to reflect a more cautious interpretation.
Author Response
Thank you for your valuable comments. All comments were responsed and all revisions were made point-by-point accordingly. Thank you for your consideration.
The detailed responses have also been submitted in Word format.
REVIEWER 2
The manuscript addresses a relevant topic. The study is well organized, with clear methodology, solid statistical analyses.
1. The retrospective design significantly limits the strength of causal inferences, yet in several places, the conclusions are presented too strongly. The authors should moderate their statements and clarify that the results reflect associations. Important confounding factors such as smoking, alcohol use, immunosuppressive medication, and social and economic status were not considered, their absence should be explicitly acknowledged as a limitation.
In accordance with the suggestion, we have carefully revised the manuscript to ensure that all interpretations are framed as associative rather than causal. Statements implying prediction or causality (e.g., “predictors,” “increased risk”) have been replaced with neutral phrasing such as “associated with” or “independent associations”
In addition, we have revised the “Limitations” to explicitly acknowledge the absence of these confounders as a study limitation.
2. It would be helpful to clarify whether the diagnosis of associated autoimmune diseases was verified by specialists or based solely on medical records of patients’ pathological history.
As stated in the Methods section, diagnoses of associated autoimmune diseases were considered valid only if the patients were under regular follow-up by the relevant specialty (e.g., endocrinology, rheumatology, neurology). Thus, all included AID diagnoses were verified by specialists and not based solely on the patients’ historical records. To improve clarity, we have slightly revised the sentence in the Methods section as follows:
“Medical records were reviewed for documented AID diagnoses, which were considered valid only if the patients were under regular follow-up and the diagnoses had been verified by the relevant specialty (e.g., endocrinology, rheumatology, neurology) according to current accepted clinical and serologic criteria. Cases with suspected or confirmed drug-induced autoimmune or rheumatic manifestations were excluded.”
3. Tables are clear, but highlighting statistically significant associations would improve readability.
To improve readability, statistically significant associations (p < 0.05) have now been highlighted in bold in all tables, with an explanatory note added below each table.
4. The abstract and conclusion are currently written in a relatively strong tone and should be revised to reflect a more cautious interpretation.
The abstract and conclusion have been revised to adopt a more cautious and balanced tone, emphasizing associations rather than causality.

Reviewer 3 Report
Comments and Suggestions for Authors
Detailed autoantibody profile for each autoimmune disease (such as rheumatoid factor and ACPA for rheumatoid arthritis, antinuclear antibody for SLE) in the study patients should be provided, in addition to authoantibodies related to thyroid disease.
All classification criteria for each autoimmune disease (such as 2010 ACR/EULAR classification criteria for rheumatoid arthritis) should be provided. And was the diagnosis of each autoimmune disease made by corresponding specialists?
Detailed information on treatment especially immunosuppresants/immunomodulators should be provided. How about the relation between administered drugs and rheumatoic -disease-like manifestations as adverse effects?
Table S1 ‘Sjögren syndrome’ has been renamed in to Sjögren’s disease.
Author Response
Thank you for your valuable comments. All comments were responsed and all revisions were made point-by-point accordingly. Thank you for your consideration.
The detailed responses have also been submitted in Word format.
REVIEWER 3
1. Detailed autoantibody profile for each autoimmune disease (such as rheumatoid factor and ACPA for rheumatoid arthritis, antinuclear antibody for SLE) in the study patients should be provided, in addition to authoantibodies related to thyroid disease.
We thank the reviewer for this valuable comment. Extended autoantibody screening (e.g., RF, anti-CCP, ENA panel) is not routinely performed in patients with autoimmune bullous diseases unless clinically indicated. On the other hand, all autoimmune disease diagnoses—including rheumatologic ones—were verified by the relevant specialists according to accepted clinical and serologic criteria, as clarified in the Methods section.
The number of patients with rheumatologic autoimmune diseases and corresponding autoantibody data (e.g., RF, anti-CCP, ENA profile) was too small for meaningful statistical analysis, and these data were not systematically available for the entire cohort. Therefore, including them could introduce selection bias and would not align with the primary aim of our study, which was to assess the prevalence and spectrum of autoimmune diseases in PV and BP compared with matched controls.
Because antinuclear antibody (ANA) results were available for a substantial proportion of the cohort, we reported ANA positivity rates in the manuscript. All autoimmune diagnoses were confirmed by the relevant specialists based on clinical, serologic, and, where applicable, histopathological findings.
For these reasons, additional autoantibody data were not included, and this limitation is now explicitly acknowledged in the revised manuscript. However, this point will be considered in future prospective studies with more comprehensive serologic data collection.
Study Limitations were revised accordingly:
“In addition, extended autoantibody data (e.g., rheumatoid factor, anti-cyclic citrullinated peptide, extractable nuclear antigen panel) were not systematically available for the entire cohort, except for certain specific AIDs such as rheumatologic diseases, and therefore could not be analyzed. Only ANA results were available for a subset of patients and were reported descriptively. Consequently, disease-specific autoantibody comparisons were not feasible; however, diagnostic accuracy was ensured, as all AID diagnoses were specialist-confirmed based on clinical and serologic criteria.”
2. All classification criteria for each autoimmune disease (such as 2010 ACR/EULAR classification criteria for rheumatoid arthritis) should be provided. And was the diagnosis of each autoimmune disease made by corresponding specialists?
Diagnoses of autoimmune diseases were verified by the relevant specialists (e.g., endocrinology, rheumatology, neurology) and were based on current accepted clinical and serologic criteria routinely used in tertiary care settings. Specific classification criteria (e.g., 2010 ACR/EULAR for rheumatoid arthritis) were not systematically recorded due to the retrospective design, but all diagnoses were confirmed by corresponding specialists, minimizing misclassification risk.
The Methods/Study design section has been revised accordingly:
“Medical records were reviewed for documented AID diagnoses, which were considered valid only if the patients were under regular follow-up and the diagnoses had been verified by the relevant specialty (e.g., endocrinology, rheumatology, neurology) according to current accepted clinical and serologic criteria. Cases with suspected or confirmed drug-induced autoimmune or rheumatic manifestations were excluded.”
3. Detailed information on treatment especially immunosuppresants/immunomodulators should be provided. How about the relation between administered drugs and rheumatoic -disease-like manifestations as adverse effects?
All patients with AIBDs were receiving or had previously received systemic corticosteroids and/or other immunosuppressive or immunomodulatory agents (e.g., azathioprine, mycophenolate mofetil, rituximab) as part of standard treatment. However, treatment data were not systematically collected for the control group, and comparative analyses regarding immunosuppressive exposure were therefore not performed. This limitation has been clarified in the revised manuscript.
All rheumatologic diseases included in the analysis were verified by the relevant specialty (e.g., rheumatology). Cases with suspected or confirmed drug-induced autoimmune or rheumatic manifestations were excluded. Cases were considered drug-induced if the treating specialist explicitly documented a temporal relationship between the onset of rheumatic manifestations and exposure to an immunosuppressive or immunomodulatory drug, or if the diagnosis was recorded as “drug-induced” in the medical report. Such cases were excluded from the analysis to minimize potential misclassification.
Revisions were made in the Methods and Study Limitations sections as follows:
Methods: “Medical records were reviewed for documented AID diagnoses, which were considered valid only if the patients were under regular follow-up and the diagnoses had been verified by the relevant specialty (e.g., endocrinology, rheumatology, neurology) according to current accepted clinical and serologic criteria. Cases with suspected or confirmed drug-induced autoimmune or rheumatic manifestations were excluded.”
Study Limitations: “Importantly, key potential confounders such as smoking, alcohol use, immunosuppressive medication, socioeconomic status, and comorbidities were not available and could not be adjusted for. All patients with AIBDs were treated with systemic corticosteroids and/or other immunosuppressive or immunomodulatory agents (e.g., azathioprine, mycophenolate mofetil, rituximab); however, treatment data were not available for the control group, precluding comparative analyses by immunosuppressive exposure.”
4. Table S1 ‘Sjögren syndrome’ has been renamed in to Sjögren’s disease.
We have replaced “Sjögren syndrome” with “Sjögren’s disease” and revised the abbreviation to SjD.

Reviewer 4 Report
Comments and Suggestions for Authors
Thank you very much for this interesting manuscript.
I have few questions and suggestions.
Materials and methods, at the end of 2.1 section, you wrote: "Thyroid autoimmunity was assessed by serum anti-thyroid peroxidase (anti-TPO) and anti-thyroglobulin (anti-Tg) levels".
In the "3.5. Comparisons of Laboratory Tests" section you wrote: "Overall anti-TPO and anti-Tg positivity did not differ significantly between BP and controls"
But in the supplermentary Table S1 you wrote differences between BP and control group:
|
AIDs |
PV group n (%) |
Control group of PV n (%) |
p-value |
BP group n (%) |
Control group of BP n (%) |
p-value |
|
HT |
15 (5.2%) |
56 (4.9%) |
0.808 |
16 (5.6%) |
31 (2.7%) |
0.014 |
In the results "3.1. Association of AIBDs with AIDs" section you wrote: "By contrast, BP had a higher prevalence of Hashimoto thyroiditis (HT)"
In the results "3.3 Stratified Analyses by Sex and Age" section you wrote: "Conversely, among males, HT was more frequent in BP than in matched controls (p = 0.011), with no difference in females"
Please, make sure that this data are correct.
Usually, women are affected of HT much more frequently than men, with prevalence rates in women ranging from 5-15% compared to 1-5% in men. Prevalence also tends to increase with age.
How can HT be the prevalence of HT be so low as reported in the Table S1 in BP matched controls?
However, you wrote contradictory sentences, such as in the 3.5 section: "Overall anti-TPO and anti-Tg positivity did not differ significantly between BP and controls"
Moreover, How could the difference written in table S1 be due only to differences among men HT prevalence, without women HT prevalence differences, as written in the results "3.3 Stratified Analyses by Sex and Age" section: "Conversely, among males, HT was more frequent in BP than in matched controls (p = 0.011), with no difference in females"
Please, clarify these inconsistencies.
Check your raw data, and change the text, results and table accordingly.
Many thanks.
Author Response
Thank you for your valuable comments. All comments were responsed and all revisions were made point-by-point accordingly. Thank you for your consideration.
The detailed responses have also been submitted in Word format.
REVIEWER 4
Thank you very much for this interesting manuscript. I have few questions and suggestions.
1. Materials and methods, at the end of 2.1 section, you wrote: "Thyroid autoimmunity was assessed by serum anti-thyroid peroxidase (anti-TPO) and anti-thyroglobulin (anti-Tg) levels".
In the "3.5. Comparisons of Laboratory Tests" section you wrote: "Overall anti-TPO and anti-Tg positivity did not differ significantly between BP and controls"
But in the supplermentary Table S1 you wrote differences between BP and control group:
|
AIDs |
PV group n (%) |
Control group of PV n (%) |
p-value |
BP group n (%) |
Control group of BP n (%) |
p-value |
|
HT |
15 (5.2%) |
56 (4.9%) |
0.808 |
16 (5.6%) |
31 (2.7%) |
0.014 |
In the results "3.1. Association of AIBDs with AIDs" section you wrote: "By contrast, BP had a higher prevalence of Hashimoto thyroiditis (HT)"
In the results "3.3 Stratified Analyses by Sex and Age" section you wrote: "Conversely, among males, HT was more frequent in BP than in matched controls (p = 0.011), with no difference in females"
Please, make sure that this data are correct.
All data were carefully rechecked, and no inconsistencies were found. Diagnoses of autoimmune thyroid diseases were verified by endocrinologists according to accepted diagnostic criteria, whereas anti–thyroid peroxidase (anti-TPO) and anti-thyroglobulin (anti-Tg) levels were evaluated as serologic markers and did not by themselves define the diagnosis.
Accordingly, while Hashimoto thyroiditis (HT) prevalence was higher among patients with bullous pemphigoid (BP), overall anti-TPO/Tg positivity did not differ significantly between BP and controls. In sex- and age- stratified analyses, HT was more frequent among males with BP compared with controls, whereas no significant difference was observed in females.
This reflects the expected discrepancy between clinical and serologic autoimmunity, which was already discussed in the manuscript. As noted in the Discussion section, the reliance on single-point antibody measurements, the distinction between serologic and clinical disease, and the use of immunosuppressive therapies in AIBDs may partly explain this divergence.
The relevant methodological clarification has been added to “Methods” section for clarity:
“Medical records were reviewed for documented AID diagnoses, which were considered valid only if the patients were under regular follow-up and the diagnoses had been verified by the relevant specialty (e.g., endocrinology, rheumatology, neurology) according to current accepted clinical and serologic criteria. Cases with suspected or confirmed drug-induced autoimmune or rheumatic manifestations were excluded. In addition, thyroid autoimmunity was assessed by serum anti–thyroid peroxidase (anti-TPO) and anti-thyroglobulin (anti-Tg) levels, which were analyzed as serologic markers and did not by themselves define the diagnosis.”
2. Usually, women are affected of HT much more frequently than men, with prevalence rates in women ranging from 5-15% compared to 1-5% in men. Prevalence also tends to increase with age.
How can HT be the prevalence of HT be so low as reported in the Table S1 in BP matched controls?
However, you wrote contradictory sentences, such as in the 3.5 section: "Overall anti-TPO and anti-Tg positivity did not differ significantly between BP and controls"
We have carefully rechecked all data and confirmed their accuracy. The low prevalence of Hashimoto thyroiditis (HT) in the BP-matched control group (2.7%) likely reflects several methodological and population-related factors rather than data inconsistency.
First, our control population was dermatology clinic-based rather than population-based. Therefore, endocrine disorders such as HT are expected to be underrepresented, as many individuals with thyroid disease are typically followed by endocrinologists and not seen in dermatology settings.
Second, only specialist-confirmed HT diagnoses were included. Participants with positive thyroid autoantibodies (anti-TPO or anti-Tg) but without a documented clinical diagnosis by an endocrinologist were not classified as having HT. This rigorous definition ensures diagnostic specificity but inevitably leads to a lower recorded prevalence compared to population surveys that rely on serologic or self-reported data.
Third, the lack of significant differences in anti-TPO/Tg positivity between BP and controls (Section 3.5) reflects the distinction between serologic and clinical autoimmunity. Serologic positivity does not necessarily indicate overt thyroid disease. As disscussed in Discussion section, the reliance on single-point antibody measurements, the distinction between serologic and clinical disease, and the use of immunosuppressive therapies in AIBDs may partly explain this discrepancy.
We have added clarifying sentences in the Methods and Discussion sections to better explain this distinction and the rationale for our diagnostic approach.
Methods:
“Medical records were reviewed for documented AID diagnoses, which were con-sidered valid only if the patients were under regular follow-up and the diagnoses had been verified by the relevant specialty (e.g., endocrinology, rheumatology, neurology) according to current accepted clinical and serologic criteria. Cases with suspected or confirmed drug-induced autoimmune or rheumatic manifestations were excluded. In addition, thyroid autoimmunity was assessed by serum anti–thyroid peroxidase (an-ti-TPO) and anti-thyroglobulin (anti-Tg) levels, which were analyzed as serologic markers and did not by themselves define the diagnosis.”
Discussion (4.2. Distinct Thyroid Autoimmunity and AIBD Profiles):
“The relatively low HT prevalence in BP controls likely reflects the clinic-based sample and the use of strict, specialist-confirmed diagnoses rather than serology-based or self-reported definitions.”
3. Moreover, How could the difference written in table S1 be due only to differences among men HT prevalence, without women HT prevalence differences, as written in the results "3.3 Stratified Analyses by Sex and Age" section: "Conversely, among males, HT was more frequent in BP than in matched controls (p = 0.011), with no difference in females"
Please, clarify these inconsistencies.
Check your raw data, and change the text, results and table accordingly.
Many thanks.
We thank the reviewer for this important comment. All data were rechecked, and found to be correct and internally consistent. While the overall prevalence of Hashimoto thyroiditis (HT) was higher in the BP group than in controls (5.6% vs. 2.7%, p = 0.014), this difference was mainly driven by the more pronounced association observed among males (6.1% vs. 1.5%, p = 0.011). Among females, HT prevalence was also higher in BP (5.3% vs. 3.5%), but the difference did not reach statistical significance (p = 0.287), likely due to smaller effect size and limited statistical power in this subgroup. Thus, the overall difference reflects a consistent trend across sexes, with a stronger effect in males rather than an exclusively male-driven association. This sex-specific pattern and possible biological explanations have been addressed in the Discussion section.
Discussion section (4.2. Distinct Thyroid Autoimmunity and AIBD Profiles) was revised accordingly:
“…Notably, HT prevalence was also numerically higher among women with BP than in controls, although this difference did not reach statistical significance, indicating a consistent but more pronounced trend in males…”

Round 2
Reviewer 1 Report
Comments and Suggestions for Authors
Thank you for your thorough response to my comments. I appreciate the clear justification provided for all Comments, and I am satisfied with the explanations you've given for several other points. Your revisions have addressed my concerns, and I am happy to recommend your manuscript for publication.
Reviewer 2 Report
Comments and Suggestions for Authors
I thank the authors for their reply and the changes made. I believe that my requests have been fully fulfilled.
Author Response
RESPONSE TO THE REVIEWERS-ROUND 2
REVIEWER 2
I thank the authors for their reply and the changes made. I believe that my requests have been fully fulfilled.
We thank the reviewer for the kind feedback and constructive suggestions that improved our manuscript.

Reviewer 3 Report
Comments and Suggestions for Authors
(No further comments)
Author Response
RESPONSE TO THE REVIEWERS-ROUND 2
REVIEWER 3
We thank the reviewer for the kind feedback and constructive suggestions that improved our manuscript.

Reviewer 4 Report
Comments and Suggestions for Authors
Thank you very much for your comments.
I only suggest to specify that diagnosis of HT in your study was performed when an endocrinologist evaluated at least a subclinical hypothyroidism.
However, I think euthyroid HT exists, but if you didn't evaluated it as HT in your study this should be clearly written.
Many thanks
Author Response
RESPONSE TO THE REVIEWERS-ROUND 2
Thank you for your valuable comments. All comments were responsed and all revisions were made point-by-point accordingly.
We also attached a Word document containing our detailed responses. Thank you for your kind consideration.
REVIEWER 4
Thank you very much for your comments.
I only suggest to specify that diagnosis of HT in your study was performed when an endocrinologist evaluated at least a subclinical hypothyroidism.
However, I think euthyroid HT exists, but if you didn't evaluated it as HT in your study this should be clearly written.
Many thanks
We appreciate the reviewer’s suggestion. The diagnostic criteria for HT have been clarified in the Methods section to indicate that only endocrinologist-confirmed cases, including euthyroid HT, were accepted.
“…In addition, thyroid autoimmunity was assessed by serum anti–thyroid peroxidase (anti-TPO) and anti-thyroglobulin (anti-Tg) levels, which were analyzed as serologic markers and did not by themselves define the diagnosis. Diagnosis of HT was accepted only when evaluated and confirmed by an endocrinologist, based on clinical or subclinical hypothyroidism with positive thyroid autoantibodies and/or characteristic ultrasonographic findings. Euthyroid HT cases confirmed by endocrinology consultation were also included.” This part has been added to “Methods/Study design” section.
